# Position: Machine Learning for Heart Transplant Allocation Policy Optimization Should Account for Incentives

**Ioannis Anagnostides** [* 1]  **Itai Zilberstein** [* 1]  **Zachary W. Sollie** [2]  **Arman Kilic** [2]  **Tuomas Sandholm** [1]

## Abstract

The allocation of scarce donor organs constitutes one of the most consequential algorithmic challenges in healthcare. While the field is rapidly transitioning from rigid, rule-based systems to machine learning and data-driven optimization, we argue that current approaches often overlook a fundamental barrier: *incentives*. In this position paper, we highlight that organ allocation is not merely an optimization problem, but rather a complex game involving organ procurement organizations, transplant centers, clinicians, patients, and regulators. Focusing on US adult heart transplant allocation, we identify critical incentive misalignments across the decision-making pipeline, and present data showing that they are having adverse consequences today. Our main position is that the next generation of allocation policies should be *incentive aware*. We outline a research agenda for the machine learning community, calling for the integration of mechanism design, strategic classification, causal inference, and social choice to ensure robustness, efficiency, fairness, and trust in the face of strategic behavior from the various constituent groups.

## 1. Introduction

The allocation of scarce donor organs constitutes one of the most consequential algorithmic problems in healthcare. Organ transplantation remains in many cases the only viable option for critically ill patients. Unfortunately, demand far outweighs existing supply: in the United States alone, more than 100,000 patients are currently on the waitlist (HRSA, 2025b). For decades, the mechanisms governing organ allocation were rigid, rule-based priority systems handcrafted primarily by medical committees composed of domain experts. However, as the complexity of patient data rapidly grows and the pressure to improve efficiency intensifies, there is a widespread push to leverage machine learning and data-driven optimization to improve decision making. Many such algorithmic approaches have already been deployed in practice (Abraham et al., 2007; Valapour et al., 2022; Mayer & Persijn, 2006; Watson et al., 2020; Papalexopoulos et al., 2023).

In order to fully reap the rewards promised by such advances, there is a fundamental, and often overlooked, barrier: *incentives*. Organ allocation is not a static optimization problem, but rather a complex *game* involving transplant centers (*i.e.*, hospitals), *organ procurement organizations (OPOs)*, clinicians, and patients, each of whom has their own objectives and responds strategically to policy fluctuations, as we document extensively in this paper. This behavior exists under rule-based policies, and will continue to influence policy design going forward. The presence of such incentives introduces challenges that the full might of supervised learning cannot hope to solve on its own. Training even the most sophisticated predictive models on historical data without accounting for concomitant strategic shifts risks creating models that fail to deliver in practice.

In this paper, we identify several incentive design failures in organ allocation, which create ample vulnerabilities for strategic manipulation by various stakeholders. We substantiate our findings through analyses of historical data from the *United Network for Organ Sharing (UNOS)*. We focus primarily on heart transplant allocation, although most of the issues we identify are present in other organs as well. Our main position is that **machine learning for heart transplant allocation policy optimization should be incentive aware**. Furthermore, machine learning and data-driven approaches have a key role to play in mitigating the adverse effects of misaligned incentives present in the current system. Going forward, we call for the proactive design of incentive structures that steer the system toward more efficient and equitable outcomes.

**Roadmap**    In the remainder of this paper, we high-

---

*Equal contribution    [1]Department of Computer Science, Carnegie Mellon University, Pittsburgh, PA [2]Department of Surgery, Division of Cardiothoracic Surgery, Medical University of South Carolina, Charleston, SC. Correspondence to: Ioannis Anagnostides <ianagnos@cs.cmu.edu>, Itai Zilberstein <izilbers@cs.cmu.edu>.

*Proceedings of the 43$^{rd}$ International Conference on Machine Learning*, Seoul, South Korea. PMLR 306, 2026. Copyright 2026 by the author(s).

light incentive misalignments across different stages of the decision-making pipeline, and identify roles machine learning can play to mitigate these vulnerabilities. Section 2 examines how clinical features, especially device utilization, are prone to manipulation to game priority tiers. Section 3 discusses the opaque and often exploited mechanism of out-of-sequence allocations. Section 4 investigates how the current performance monitoring regime can adversely affect the decision making of transplant centers. Section 5 addresses inequities in waitlist access, namely multi-listing and gatekeeping of waitlist admission. On the policy design level, Section 6 critiques the way preferences were elicited to establish community priorities in previously deployed policies, and suggests better approaches.[1]

## 2. Patient Feature Manipulation

The current heart allocation rule in the US divides patients into 6 tiers based on estimated medical urgency. It was deployed in 2018 to create a more granular separation *vis-à-vis* the previous 3-tier system (Kilic et al., 2021).

**The device game** The status assigned to a potential recipient—which dictates the likelihood of a donor match—is a proxy for medical urgency, but crucially depends on *device utilization*. This creates an opportunity for clinicians to alter patient features in order to inflate medical severity and thereby boost the chance their patient will be matched. We coin this the *device game*.

There is considerable evidence documenting that this type of manipulation is rampant. Let us discuss an illustrative example. Under the 2018 policy update, patients bridged with *intra-aortic balloon counterpulsation (IABP)* were designated status 2, thus giving them a relatively higher priority (Huckaby et al., 2020). Conversely, patients supported with *durable left ventricular assist devices (LVADs)* have reduced priority to status 4; compared to the prior allocation policy, this is a lower priority. The proportion of patients bridged with an IABP increased significantly following the adoption of the new policy from $7.0\%$ to $24.9\%$ (Huckaby et al., 2020); this is more than a three-fold increase. The concern is that the policy change triggered a shift in the selection of recipients for bridging with IABP (Khazanie & Drazner, 2019). Notwithstanding the potential benefits and legitimate medical reasons, using IABP has certain adverse effects, such as the potential for visceral malperfusion (Huckaby et al., 2020). Hemorrhagic complications have also been associated with longer duration of IABP support (Valente et al., 2012; Boudoulas et al.,

2014). While clinical practice has shifted toward more advanced micro-axial flow pumps, specifically the Impella 5.5 (Ramzy et al., 2021), the incentive structure and strategic dilemma remain intact.

Goodhart's law succinctly explains the fundamental problem: "When a measure becomes a target, it ceases to be a good measure (Goodhart, 1975)."

**Policy optimization going forward** There are growing calls to move away from the current 6-tier system as it fails to sufficiently account for pretransplant and post-transplant survival (Zhang et al., 2024). For example, in lung transplant allocation, the framework of *continuous distribution* was recently deployed in the US, and is currently under review to deploy for heart allocation (Papalexopoulos et al., 2023). As we discuss further in Section 6, continuous distribution is based on a scoring system comprising multiple factors, yielding a continuous (*i.e.*, real-valued) priority for each patient. This mitigates cliff-edge effects of tier boundaries, where a marginal change in therapy leads to a considerable priority upgrade. Even so, misreporting of the form highlighted earlier remains possible since the underlying classification/regression algorithms upon which continuous distribution relies are prone to feature manipulation.

**Connection to strategic classification** From a broader standpoint, feature manipulation has recently garnered considerable attention in the intersection of machine learning and game theory, and is known as *strategic classification* (Hardt et al., 2016; Chen et al., 2020; 2018; Dong et al., 2018). A canonical motivating application in that framework is predicting credit default risk. Certain features can be strategically misreported in order to elevate the chance of being classified favorably. There is a cost to misreporting (such as installing a suboptimal device), especially when the reported feature is far from the ground truth, so optimal manipulation balances between two conflicting objectives. Figure 1 illustrates this issue in the context of heart transplantation.

A common heuristic employed in such settings to deal with the distribution shift that results from strategizing is to perform *repeated* risk minimization, which is known to converge under certain assumptions (Perdomo et al., 2020). We argue that future work in transplantation needs to leverage ideas developed in the strategic classification literature to combat incentive misalignments. Adapting this framework to survival analysis presents distinct research challenges. Furthermore, *causal inference* (*e.g.*, Imbens & Rubin, 2015; Peters et al., 2017) can be used to discern features that have a causal link to medical urgency.

Feature manipulation is not just about device utilization, but has broader manifestations. An example of this concerns the amount of time a patient spends on the waitlist.

---

[1]Throughout this paper, we use the term "policy" to broadly encompass the entire institutional mechanism, not just the allocation rule. We envision the development of multiple ML components that need to be co-designed so as to build incentive-aware policies.

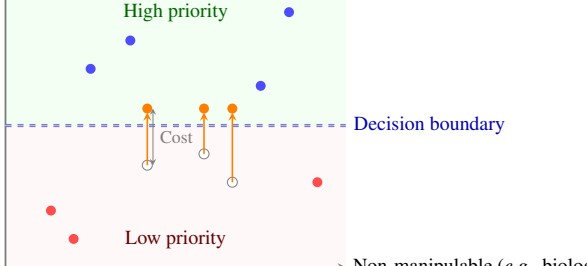

*Figure 1.* Strategic classification in organ allocation. Patients below the threshold (red region) incur a cost to manipulate certain features (*e.g.*, device implantation) to cross the decision boundary and gain high priority status (green region), gaming the classifier.

The current allocation policy prioritizes patients who have been on the waitlist longer, at least without a status change. This creates a possible incentive for centers to list patients early, perhaps earlier than medically warranted.

Current policies that prioritize wait time, combined with broad status tiers susceptible to feature manipulation, severely impact patient outcomes. Table 1 presents mortality and transplant metrics for the highest-urgency candidates, revealing a system operating on razor-thin margins. The median time to transplant (26 days) precedes the median time to death (36 days) by a margin of only 10 days. Furthermore, nearly 14% of waitlist deaths occur within the first week. Because these patients deteriorate too fast to accrue priority based on wait time, they are disadvantaged by a system that rewards stable candidates who list early to bank time. The immense variability in survival times—indicated by an interquartile range of 13 to 118 days—suggests that the single status 1 classification conflates patients at immediate risk with those who are more stable. This reinforces the need for continuous priority functions over using a few coarse buckets and are robust to manipulation.

What can be done to combat feature manipulation? One

| Metric | Value |
| --- | --- |
| **Mortality timing** | |
| Deaths within 3 days of listing | 6.5% |
| Deaths within 7 days of listing | 13.7% |
| **Median times** | |
| Time to transplant | 26 days |
| Time to death | 36 days |
| **Variability** | |
| Time to death interquartile values | 13 − 118 days |
| (*i.e.*, 25th & 75th percentile) | |

*Table 1.* Highest-urgency (status 1A pre-2018, status 1 post-2018) US adult heart transplant candidate outcomes (2010–2024).

remedy is to build machine learning models that rely less, or at least make more judicious use of, manipulable features, such as device utilization and accrued wait time.

> **Recommendation:** Evaluating medical urgency should be based more on non-manipulable features. Model training should be incentive aware.

Another way to mitigate the problem is to increase the cost of manipulation. One feasible way could be through the use of *randomized audits* (Townsend, 1979; Mookherjee & Png, 1989). In mechanism design, this is called *selective verification* (*e.g.*, Fotakis et al., 2016; Zhang et al., 2021), and is known to be remarkably effective in aligning incentives even with a small number of verifications. We caution that poorly designed interventions—such as audits—could have unintended consequences, and designing effective guardrails is an important avenue for future research.

> **Recommendation:** Randomized audits can be introduced. A penalty should be imposed if a board ascertains that device usage cannot be clinically justified.

The current system already requires clinicians to justify device usage with evidence based on physiological measurements (OPTN, 2023a), but we argue that it still leaves too much room to play the device game.

## 3. Open Offers and Out-of-Sequence Allocation

While the US allocation algorithm is designed to be a rigid priority queue based on factors such as medical urgency and geographic proximity, in practice, it operates alongside a discretionary mechanism known as *out-of-sequence allocation* (Clerkin et al., 2025; Benkert et al., 2025).

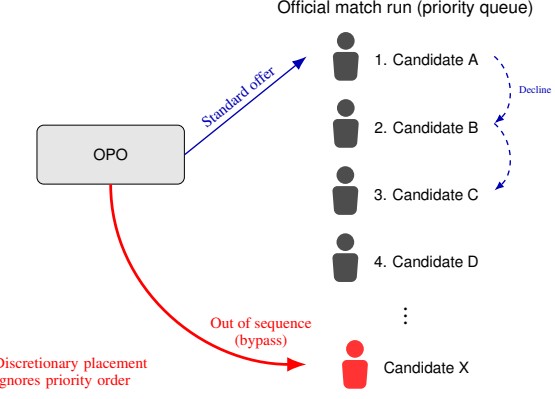

*Figure 2.* Out-of-sequence allocation. While the standard policy (blue) mandates sequential offers starting from the highest priority, open offers (red) allow OPOs to bypass the entire queue and allocate the organ directly to a lower-priority candidate.

In particular, the official policy authorizes the organ procurement organizations (OPOs)—that is, the regional organizations that conduct the allocation of deceased donor organs—to bypass the priority queue and allocate organs directly to a specific transplant center (*i.e.*, hospital) if the OPO ascertains that there is a risk of organ loss due to excessive ischemic time. Organs deteriorate with the passage of time (Barah et al., 2022). Therefore, traversing a long list of offers and offer rejections to possibly hundreds of potential recipients can result in a significant time delay; issuing even a single offer requires considerable logistical effort and precious time. To accelerate this process, OPOs may resort to *open offers*, broadcasting the organ's availability to multiple centers simultaneously.

Although this pathway of out-of-sequence allocation afforded to the OPOs may be necessary in certain instances to prevent organ wastage, the threshold to trigger it is opaque and subjective. Unlike the allocation policy, the rules of which are entirely transparent, the out-of-sequence mechanism relies on the judgment of individual OPOs.

Why would an OPO choose to bypass the priority queue even when there is no medical urgency to do so? One reason is that OPOs are incentivized to maximize the number of organs placed while minimizing logistical effort. Out-of-sequence allocation can be significantly more efficient than making sequential offers in a short time window. Another factor is the center's relationship with the OPO, creating conditions under which collusion can emerge.

Since 2021, OPOs have officially come under scrutiny about wasting organs and have been more closely monitored regarding their organ wastage on an OPO-by-OPO basis (Centers for Medicare & Medicaid Services, 2020). This has led the OPOs to use out-of-sequence offers dramatically more prevalently. A recent investigation by the New York Times brought to light the alarming number of open offers (Rosenthal et al., 2024), describing the current organ transplant system as being in "chaos." Although out-of-sequence allocation was a rare exception in the past (on the order of $1\%$), the number of out-of-sequence allocations has skyrocketed. For kidneys, the rate rose from $2\%$ in 2020 to $18\%$ in 2023 (Mohan et al., 2025). Across all organs, approximately $19\%$ of allocations in 2023 were classified as out of sequence (Rosenthal et al., 2024). While out-of-sequence allocations are less frequent in heart transplantation, their occurrence was on a steady upward trend also (HRSA, 2025a). At the same time, the number of discarded organs has also increased, casting doubt on the claim that such discretion is necessary to prevent organ discard. Another concerning finding by Rosenthal et al. (2024) is that out-of-sequence allocations predominantly favor certain demographic groups, typically more affluent individuals (Mohan et al., 2025; Corbie et al., 2025).

The remedy to mitigate out-of-sequence allocations is to bring more transparency into the decision making, so that OPOs cannot exploit this option for reasons other than medical necessity. Furthermore, should an open offer be deemed the best course of action, it should still adhere to a regional sub-policy that is transparent—for example, broadcast to all centers within a certain distance—rather than allowing direct, manual placement to select centers.

> **Recommendation:** The decision to switch from the priority queue dictated by the policy to open offers should not be discretionary. Rather, it should be triggered by hard constraints; *e.g.*, ischemic time exceeding a certain threshold together with other factors.

Recent developments demonstrate that OPOs do respond rapidly to changes in policy and monitoring. Following federal scrutiny of OPOs' out-of-sequence allocations starting in 2025 (HRSA, 2025a), the rate of such exceptions plummeted from 20% in 2024 to 9% by early 2026 (Rosenthal, 2026).

Machine learning has many key roles to play in this. It could be used to determine the optimal threshold to trigger an out-of-sequence allocation and to which center or patient the organ should be allocated. This should be based on real-time indicators of the donor organ's state. Computer vision techniques could be employed during *ex-vivo* perfusion (*cf.* Bello et al., 2019) to evaluate this. This viability assessment should be coupled with a dynamic evaluation of the recipients in the local region to ascertain whether there is a viable out-of-sequence offer. Furthermore, the overall national policy should be optimized (automatically), taking into account the fact that such regional sub-policies will be used. To our knowledge, this critical problem remains essentially entirely unexplored.

## 4. Strategic Offer Rejection and Performance Monitoring

Once an offer is issued—whether in accordance with the national policy's priority queue or an out-of-sequence offer—the transplant center has the option of rejecting that offer. This decision is made by a medical committee that weighs whether the proposed match is viable. Now, the reality is that centers have their own distinct incentives, and there are concerns that centers may end up rejecting viable offers or accepting offers that are not the best for the patient in expectation, contributing to system inefficiency (including organ wastage in the case of rejected offers).

We argue that a primary force driving this behavior is the performance monitoring regime. Transplant centers are evaluated semi-annually by the *Scientific Registry of Transplant Recipients (SRTR)*. These program-specific re-

ports (Scientific Registry of Transplant Recipients, 2025b) publicly rate centers on a 5-tier system (Scientific Registry of Transplant Recipients, 2025a). The evaluation is based on three key metrics: survival on the waiting list (pretransplant mortality rate), time to transplant (transplant rate), and 1-year graft (*i.e.*, the transplanted organ) survival. The result is that centers operate under the constant pressure of regulatory flagging; a poor evaluation can lead to loss of insurance contracts or even revocation of certification.

**Pitfalls in current performance monitoring** Although performance monitoring can increase accountability and drive continuous quality improvement, it is unclear whether the current system achieves those goals. There are concerns that the current regime incentivizes risk aversion. From the center's perspective, accepting an organ poses an asymmetric risk: a successful operation results in a marginal gain in volume, but an unsuccessful one can disproportionately undermine the center's metrics. While SRTR does use an estimated score function to adjust the evaluation for different levels of risk in different cases, it is not clear whether it is sufficient; if not, there is an incentive to admit easier cases and not difficult ones. The existing evaluation system may also not accurately reflect the key metrics of interest. For example, focusing on 1-year organ survival disregards longer-term outcomes, which are arguably more informative when it comes to determining the efficiency of the system. The key reason behind the prevalent use of short-term metrics, such as 1- or 3-year outcomes, is practical: they provide timely feedback necessary for rapid performance evaluation. However, this may be at odds with the objective of maximizing total life years gained.

These pressures manifest unevenly across transplant centers. As shown in Table A1, small centers receive only a fraction of the offers that larger centers receive. In turn, small centers generally suffer higher waitlist mortality and accept offers at a rate nearly 50% higher than large centers. Conversely, large centers operate in an environment of offer abundance and have the luxury of being highly conservative, rejecting the vast majority of organs while waiting for lower-risk matches.

**Evaluation cycle gaming** Moreover, the temporal nature of evaluations may distort incentives. Data reporting windows close in April and October (Scientific Registry of Transplant Recipients, 2025c). Horizon effects can force centers to become hyper-conservative as the reporting window closes in order to protect their current standing. Conversely, a center with a guaranteed top-tier evaluation may be inclined to take more risks in order to conduct more transplants and thus to make more profit. Such distortion is not unique to organ allocation. For example, agents altering their risk tolerance as the reporting horizon comes to a close is well-documented in financial markets—becoming

risk-averse to lock in a superior standing or risk-seeking to gamble for resurrection (Chevalier & Ellison, 1997). Similar gaming is common in sports toward the end of a match.

We examine this hypothesis using UNOS heart transplant data, summarized in Figure 3. Interpreting these patterns requires nuance, as they are influenced by a confluence of biological and logistical factors unrelated to policy. For instance, the surge in organ supply observed in December and January (Figure 3a) aligns with seasonal spikes in cerebrovascular mortality due to colder climates (Lichtman et al., 2013). Similarly, fluctuations in transplant volume can reflect human resource constraints, such as reduced surgical staffing during holiday periods. However, distinct patterns emerge around the spring reporting deadline that lack clear clinical drivers. While acceptance rates remain universally low ($< 1\%$), we observe a statistically significant spike in transplant volume and acceptance probability in May, immediately following the April deadline (Figures 3b, 3c). The absolute magnitude of this volatility is small, but its timing is consistent with deferring riskier procedures to the start of a new reporting cycle, where there is a longer horizon to recover if transplant failure occurs. A more rigorous causal analysis remains necessary to definitively attribute this spike to strategic horizon effects rather than unobserved clinical or logistical variation.

The disconcerting reality is that an overwhelming proportion of offers end up being rejected. Offer rejection triggers a vicious cycle where delays degrade organ quality, making subsequent rejection and ultimate discard more likely. The high organ discard ratio may be exacerbated by the current evaluation system.

To be clear, we are not arguing against any performance evaluation. Rather, we emphasize the need to refine current evaluation mechanisms to mitigate unintended consequences. Machine learning can empower the development of more accurate risk adjustment models in particular, and better evaluation systems in general. Also, moving away from the rigid semi-annual evaluation system to a continuous monitoring system (*cf.* Axelrod et al., 2006) would reduce the incentive to game the evaluation cycle. As in all parts of the pipeline, interpretability is essential for mechanism design guarantees to confer tangible benefits; if evaluation rules are opaque "black boxes," clinicians may revert to suboptimal practices out of distrust (Rudin, 2019; Ahmad et al., 2018; ElShawi et al., 2021; Stiglic et al., 2020).

> **Recommendation:** The performance evaluation system should be refined to mitigate risk aversion, horizon effects, and organ wastage.

The fact that there are so many rejections can also be attributed to a poor priority queue created by the existing policy. Viewed from that perspective, one can argue that offer

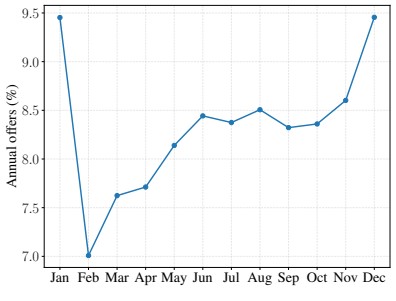

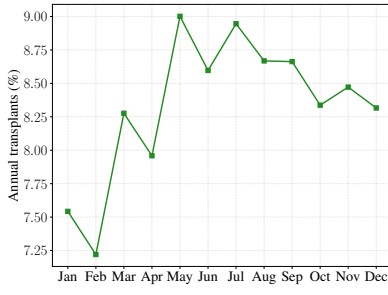

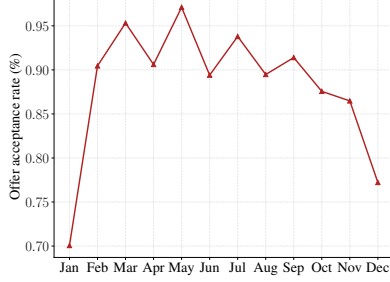

*(a)* Distribution of annual offers.

*(b)* Distribution of transplants.

*(c)* Offer acceptance rates.

*Figure 3.* Distribution of annual accepted offers, heart transplants performed, and center acceptance rates per month among adult patients (2010–2024). Panel (a): distribution of offers. Panel (b): distribution of transplants performed. Panel (c): offer acceptance rates.

rejections serve to correct the inefficiency of the global policy. If this is the case, improving the allocation algorithm would have the concomitant effect of moderating offer rejections. Encouragingly, a wide array of emerging machine learning and data-driven algorithms are poised for deployment to address these challenges. Going forward, one approach, whose implementation would admittedly face considerable obstacles, is to remove the option from centers to reject offers. Recent work suggests that enforcing acceptance can improve overall efficiency (Anagnostides et al., 2025), *subject to using improved allocation policies*. A more realistic alternative would be to penalize excessive turndowns of viable organs, or, conversely, to create credit score systems that incentivize centers to accept more offers. Similar ideas have found fertile ground in the context of kidney exchange to encourage centers to truthfully report patients and donors (*e.g.*, Hajaj et al., 2015).

> **Recommendation:** In parallel with optimizing the global allocation policy, credit score systems can be developed to limit offer rejections.

## 5. Strategic Listing and Delisting

Strategic incentives can also distort the composition of the waitlist itself, influencing both the initial selection of candidates and the preemptive delisting of high-risk patients.

**Pre-listing selection** Before a patient qualifies to enter the waitlist, and thus becomes subject to SRTR monitoring, they must be admitted by a selection committee internal to the center. This decision relies, in part, on subjective criteria. For one, the performance monitoring regime described in Section 4 creates an incentive for centers to curate their waitlist admissions so as to maximize survival rates and minimize waitlist mortality. This pre-listing selection process is effectively invisible to the regulators.

**Delisting** Conversely, centers have an incentive to preemptively *delist* quickly deteriorating patients by designating

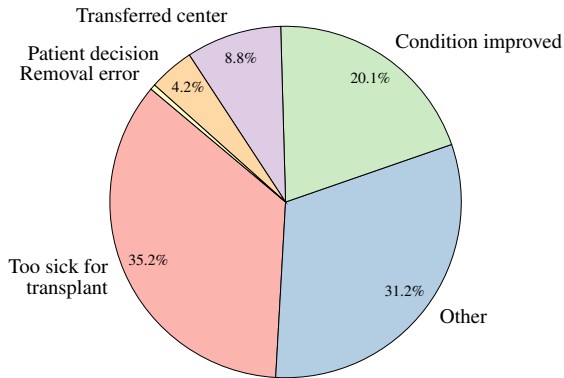

*Figure 4.* Reported center removal reasons for alive and non-transplanted US adult heart transplant candidates (2010–2024).

them "too sick for transplant." In this way, the center can prevent the death from being attributed to its program performance by removing high-risk candidates before a terminal event occurs. Unlike pre-listing selection, delisting is an explicit outcome measurement in the UNOS system. This type of strategic behavior constitutes a form of data censoring: it can distort the center's metrics, and potentially deny candidates a chance at a life-saving transplant.

Among UNOS heart transplant candidates from 2010 to 2024, we find that approximately 20% are removed from the waitlist for reasons other than transplant or death. Figure 4 illustrates the distribution of removal reasons among these patients. More than a third of these candidates were removed because the center deemed them too sick, while nearly another third were removed under the generic category of "Other." The latter category captures any patient not fitting into designated codes, rendering the specific reasons for removal in this group effectively unverifiable. We highlight this opacity as a major limitation of the current reporting, as it precludes a factual determination of the fate of these individuals.

To mitigate ambiguities and adverse effects, we argue that

the field must transition toward objective and transparent criteria for both listing and delisting. The machine learning community can contribute, for example, by developing standardized risk models that estimate the counterfactual outcomes of rejected or delisted candidates.

> **Recommendation:** Objective criteria supported by machine learning risk stratification should be introduced for waitlist admission and removal.

**Multi-listing** Multi-listing occurs when candidates register at multiple centers, exploiting regional supply disparities or center acceptance criteria to increase transplant probability. While permissible, this creates severe equity concerns by correlating access with the ability to afford duplicate evaluations and travel (Ozminkowski et al., 1997)—often via private aviation in order to be able to get to the center fast enough once an organ becomes available (Ata et al., 2017). This stratification is starkest in *transplant tourism*, where wealthy foreign nationals multi-list in the US. International patients pay premium rates—typically millions of dollars—making them financially attractive to centers seeking to maximize revenue (Rosenthal & Hansen, 2025). Multi-listing undermines the egalitarian values the allocation policy was designed to uphold.

Our analysis of UNOS adult heart transplant data from 2010 to 2024 reveals the effect of multi-listing (Table A2). While a niche practice, employed by only 2.16% of patients, multi-listed candidates achieve a disproportionately higher transplant rate (80.44% vs. 73.06%). Notably, the median wait time drops sharply once a second listing is activated. The mean distance between centers for multi-listed candidates is 379 nautical miles (NM), roughly the distance between Boston and Washington, D.C., though the maximum exceeds 2,200 NM, representing cross-country listings. We observe a clear relationship between distance and outcomes (Table A3). As the geographic spread between centers increases, transplant rates rise. Access to entirely different donor pools provides a substantial allocation advantage. These patients also exhibit a paradoxical clinical profile: they often hold higher urgency status yet suffer lower waitlist mortality, despite enduring longer wait times. Prior work also shows that multi-listing is associated with younger, white, and college educated candidates among other factors (Mooney et al., 2019).

Transitioning to a single-entry mechanism would neutralize the inequitable advantage of multi-listing. However, multi-listing involves a tradeoff between fairness and efficiency. There may be cases—such as urgent or hard-to-match patients—where multi-listing improves the overall outcomes of the system. On a technical level, counterfactual modeling and predictive systems can be used to quantify when and where multi-listing can benefit the system.

# 6. Value Aggregation and Strategic Preference Reporting

Especially as organ allocation shifts away from rigid, tier-based classification systems, a key challenge is to determine the relative importance of different objectives that may be at odds with each other. There is no definite, single objective to optimize in organ allocation, and resolving these tradeoffs requires navigating complex preferences and ethical considerations (OPTN, 2015; Keswani et al., 2025). Aligning prioritization with collective values requires robust feedback from the community. To translate abstract values into concrete parameters, policymakers have turned to preference elicitation in the past. However, this feedback process is not impervious to manipulation. Moreover, a poorly designed preference elicitation mechanism can distort community values.

This section discusses potential vulnerabilities in this crucial stage of policy design. One example is the specific methodology used by the OPTN to develop the policy currently deployed for lung allocation, and which is currently in the development phase for hearts. The preference elicitation method used there is the *analytic hierarchy process (AHP)* (Saaty, 1977).

**Continuous distribution and AHP** The continuous distribution framework (Papalexopoulos et al., 2023) determines a candidate's priority by computing a composite score, which is just a weighted sum over a set of attributes. The fundamental challenge is to select the constituent attributes and calibrate their respective weights. AHP was used to address this challenge and determine prioritization of attributes through community input. Section A provides an excerpt from the official release (OPTN, 2023b) detailing the preference elicitation methodology underpinning continuous distribution.

**Preference aggregation and elicitation** The aggregation of diverse preferences lies at the heart of social choice theory, a field that has also flourished in the AI community. However, this undertaking faces a daunting obstacle: preference aggregation is notoriously prone to strategic manipulation. The celebrated Gibbard–Satterthwaite theorem (Gibbard, 1973; Satterthwaite, 1975) establishes that any reasonable voting rule is susceptible to manipulation.

We now examine how such theoretical obstacles manifest themselves in organ allocation. As discussed above, preferences are extracted from a diverse group of stakeholders, including transplant surgeons, patients, donor families, and administrators. Although this approach appears democratic and participatory, stakeholders often have conflicting objectives that can skew their indicated preferences. In particular, reported preferences may not truthfully indicate a participant's ethical priorities.

To give a concrete example, centers have an incentive to maximize their transplant volume while maintaining high enough performance metrics to satisfy regulatory standards. As a result, the stakeholders associated with a small rural center would benefit from broader sharing, that is, lifting the geographic constraints. The opposite is true for urban centers. The consequence of this tension is that when stakeholders from such institutions engage in an AHP exercise, their responses are likely to be skewed away from an impartial ethical stance.

Similarly, candidates and their families have incentives to inflate the importance of specific attributes that would maximize their personal probability of a match. A recent tentative survey produced a weight of 13.9% for "prior living donor" status (Cummiskey et al., 2025). For a given static patient pool it seems that the weight should be zero, suggesting that factors other than medical urgency are at play. On the other hand, a positive weight incentivizes donations (of other organs than hearts from living donors), which can serve to expand the future donor pool.

Strategic manipulation is not confined to the calibration of weights, but extends to the identification of the attributes themselves. Consider, for instance, the choice between prioritizing 1-year versus 5-year post-transplant outcomes. Centers will naturally advocate for the time horizon that portrays their performance in the most favorable light.

Beyond the susceptibility to strategic manipulation, the current framework has other important limitations. First, the feature space in organ allocation is high-dimensional; a comprehensive set of pairwise comparisons would impose a prohibitive cognitive burden on stakeholders. Second, AHP relies on the "all else being equal" assumption. However, this assumption is often clinically unsound, as key factors are frequently correlated, making it impossible to meaningfully evaluate them in isolation.

> **Recommendation:** Mechanism design should guide preference elicitation and value aggregation in organ allocation policy optimization going forward.

Applying such techniques to transplant allocation requires aggregating conflicting reward functions from different stakeholders. One approach that could be beneficial here is *frugal preference elicitation from multiple parties*, which started in the context of combinatorial auctions (Conen & Sandholm, 2001). It turns out that given what the agents have expressed about their preferences so far, only some aspects of each agent's yet-unrevealed preferences need to be collected, even to find the provably optimal policy (Sandholm & Boutilier, 2006). Techniques from query learning can be applied to this setting (Zinkevich et al., 2003; Blum et al., 2004; Lahaie & Parkes, 2004). The burgeoning area of *reinforcement learning from human feedback*

*(RLHF)* (Kaufmann et al., 2024) can also provide a natural framework for addressing aggregation challenges (Conitzer et al., 2024).

**Conflating the means with the ends** Another conceptual issue of previous preference elicitation methodologies in transplantation is the failure to distinguish *means* from *ends* (Dickerson & Sandholm, 2015). Community feedback should focus on normative objectives (the "ends"). Asking stakeholders to assign weights to attributes (which are part of the "means") conflates the teleological with the instrumental, burdening human judgment with a complex optimization task that is better suited for sophisticated algorithms. Likewise, asking stakeholders to choose between two patients fails to capture the long-term distributional consequences of a policy (another part of the "means"), which should be optimized computationally.

> **Recommendation:** Preference elicitation should be about the ends, not the means. The means should be optimized computationally using data.

# 7. Alternative Views

While we maintain that incentive misalignments lead to allocation inefficiencies, we acknowledge two counterarguments: one arising from clinical reality, and another concerning the practical limitations of mechanism design.

A clinical expert may argue that what we characterize as manipulation is a necessary exercise of clinical judgment acting as a safety valve. Current risk metrics, such as status tiers, are imperfect approximations of patient urgency. They cannot capture unquantifiable or subtle cues that a seasoned clinician recognizes but an algorithm misses. Upgrading a patient's status by manipulating device utilization may not be a nefarious act of gaming, but a corrective action to counteract an imperfect policy. If we replace human discretion with data-driven models, we risk stripping the system of its ability to handle edge cases. There is a legitimate concern that algorithms, no matter how advanced, may fail to capture the full nuance of clinical reality, making human discretion a vital component of any policy.

Critics may also challenge the applicability of mechanism design in this domain. First, there are results as to what mechanism design cannot accomplish, such as the Gibbard-Satterthwaite theorem mentioned earlier and impossibility results in kidney exchange about incentivizing centers to reveal all donors (Roth et al., 2005b; Ashlagi et al., 2015; Toulis & Parkes, 2011). Second, there may be incentives that the players (*e.g.*, OPOs) have that the mechanism designer does not know about and may therefore not properly address. Third, a mechanism's incentive compatibility may become moot in practice if agents cannot comprehend it and revert to heuristic gaming simply out of a lack of trust.

These legitimate concerns highlight the need for extensive evaluation prior to deployment—a process that proved essential for building clinical trust before the implementation of kidney exchange algorithms. Modern kidney exchanges leverage complex matching algorithms based on search and integer programming—the details of which are not understandable by clinicians—over older, fully understandable dispatch rules. So, it is not always the case that the adopted solution is the more understandable one. Sometimes the community chooses efficiency over explainability, and kidney exchange is a prime example of that. Moreover, prior experience in mechanism design indicates that strong theoretical guarantees translate to robust and reliable performance for incentive-aware real-world systems. Spectrum auctions are a notable example in which well-designed mechanisms led to considerable improvements in efficiency (*e.g.*, McAfee & McMillan, 1996; Milgrom, 2004).

## 8. Conclusions and Future Research

The transition from rule-based priorities to data-driven optimization represents a paradigm shift in organ allocation. However, algorithmic sophistication alone is insufficient. Treating allocation as a learning and optimization problem fails to account for the strategic behavior of agents operating within the system.

In this paper, we laid out many problems that are prevalent in today's transplant system, with a special focus on hearts. The path forward demands incentive-aware policy optimization. We made many directional suggestions as to how this should and should not be done, giving rise to a rich and urgent research agenda. Machine learning and optimization will play key roles in automated policy design.

The recent erosion of public confidence in the current deceased-donor organ allocation system led to a significant drop in registered donors (Rosenthal, 2026). This underscores another reality: aligning incentives is not just about increasing efficiency and equity; it is also the safeguard of public trust. In a system reliant on altruistic donors, a loss of trust is a loss of supply. Ultimately, saving more lives requires not just better algorithms, but better games.

## Acknowledgments

Tuomas Sandholm and his PhD students Ioannis Anagnostides and Itai Zilberstein are supported by NIH award A240108S001, the Vannevar Bush Faculty Fellowship ONR N00014-23-1-2876, and National Science Foundation grant RI-2312342. Itai Zilberstein is also supported by the NSF Graduate Research Fellowship Program under grant DGE2140739. Arman Kilic is supported by NIH RO1 grant 5R01HL162882-03 which contributed to the funding for completion of this project. Arman Kilic is a speaker and consultant for Abiomed, Abbott, 3ive, and LivaNova, and founder and owner of QImetrix. All additional authors have no financial relationships to disclose. Any opinions, findings, and conclusions or recommendations expressed in this material are those of the author(s) and do not necessarily reflect the views of the funding agencies. We are indebted to Carlos Martinez from UNOS for answering numerous questions.

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

# A. Further background and related work

**Kidney exchange**   Incentives have been extensively explored in the related problem of *kidney exchange*. It has been documented that the kidney exchange market suffers from market failures that substantially undermine the number of transplants (Agarwal et al., 2019; Stewart et al., 2013). In particular, the kidney exchange market is largely fragmented, as most transplants are orchestrated by hospitals rather than national platforms. A major challenge there is that hospitals may sometimes withhold their easy-to-match pairs and only show hard-to-match ones to the central pool, resulting in a *free riding* problem (Ashlagi & Roth, 2014). The basic issue is that a large hospital can benefit from performing its own internal exchanges (Roth et al., 2005a). A theoretical characterization of the efficiency loss was established by Toulis & Parkes (2011). From an algorithmic perspective, Dickerson et al. (2012) demonstrate that combining a centralized dispatch with data-driven policies can greatly improve long-term efficiency in kidney exchange.

Agarwal et al. (2019) advocate for the imposition of new mechanisms and reward structures to incentivize hospitals to truthfully report patients and donors. Hajaj et al. (2015) propose the use of credits to guarantee truthful disclosure of donor-patient pairs from the transplant centers. The mechanism design problem of incentivizing hospitals to report truthfully was also addressed by Ashlagi et al. (2015), who provided certain impossibility results together with a randomized strategyproof mechanism that delivers at least half of the optimal welfare, and more recently by Blum & Gölz (2021). Market failures are less explored and understood in other organs, which present a different set of challenges.

**Signaling perspective on the device game**   From an economics standpoint, the inclusion of device usage in the allocation policy effectively treats clinical interventions as *signals* of patient severity. In the standard signaling model (Spence, 1973), for a signal to be credible and sustain a *separating equilibrium*—a state in which different types of agents (here, truly urgent versus less urgent) select different actions—the cost of sending the signal must be sufficiently high so that less urgent agents are deterred from mimicking urgent agents. More concretely, in our context, the cost corresponds to the medical risks associated with IABP support (such as vascular complications, bleeding, immobility) and other such interventions that are, to a certain extent, discretionary. The 2018 policy appears to have inadvertently induced a payoff structure in which the expected utility of the priority boost—upgrading from status 4 to status 2—outweighs the expected cost of the medical risk for a certain cohort of patients. The apparent consequence is that we ended up with, to use the economics terminology, a *pooling* or *semi-separating equilibrium* in which certain patients are strategically treated with high-risk devices to secure organ offers.

**Multiple listing**   A study of US lung transplant waitlist candidates found that although multiple listing is relatively rare, occurring in only 2.3% of candidates, it significantly impacts outcomes and highlights sociodemographic disparities (Mooney et al., 2019). Candidates who multi-list are associated with an increased likelihood of lung transplant. Our findings for multi-listed heart transplant candidates align with earlier outcomes reported from 2000–2013 (Givens et al., 2015).

**Transplant tourism**   Current policy permits foreign citizens to multi-list in the US; this is referred to as *transplant tourism* (Rosenthal & Hansen, 2025). This has given rise to instances where wealthy international patients bypass domestic candidates on the waitlist. For example, a recent investigation revealed that a Japanese national paying out-of-pocket received a heart transplant within just three days of listing in the US (Rosenthal & Hansen, 2025).

**Analytic hierarchy process**   In the context of continuous distribution, the analytic hierarchy process (AHP) was used to identify the constituent attributes and calibrate their weights. AHP is a common methodology introduced by Saaty (1977), which can be adapted to elicit preferences from pairwise comparisons such that they are compatible with established value theory (Salo & Hämäläinen, 1997).

To better explain the underlying preference elicitation methodology, we present a quote from the official release (OPTN, 2023b):

> "Anyone can participate in the AHP exercises for continuous distribution, which will occur for all organ types. In an AHP exercise for continuous distribution, participants are shown pairs of attributes (blood type vs. distance, for example) that will be used to prioritize candidates. The AHP participant must decide, if all else is considered equal, which of the two attributes is more important than the other when prioritizing a candidate for an organ."

## B. Omitted tables and figures

| Center category | Centers $N$ | Waitlist mortality (%) | Total offers | Acceptance rate (%) | $P$-value $(\chi^2)$ |
|---|---|---|---|---|---|
| *New listings/year* | | | | | |
| Small (< 10/yr) | 97 | 6.94 | 96 502 | 1.25 | |
| Medium (10–30/yr) | 48 | 5.12 | 992 940 | 0.94 | < 0.001 |
| Large (> 30/yr) | 55 | 6.01 | 3 212 332 | 0.85 | |
| *Transplants/year* | | | | | |
| Small (< 10/yr) | 111 | 5.69 | 267 595 | 1.08 | |
| Medium (10–30/yr) | 61 | 5.99 | 2 196 648 | 0.77 | < 0.001 |
| Large (> 30/yr) | 28 | 5.66 | 1 837 531 | 0.98 | |

*Table A1.* Adult heart transplant offer acceptance and waitlist mortality by center size in the US (2010–2024).

| | Single-listed | Multi-listed | | |
|---|---|---|---|---|
| **Metric** | **1 center** | **Total (2+)** | **2 centers** | **3+ centers** |
| Patients ($N$) | 49,013 (97.84%) | 1,084 (2.16%) | 1,065 (2.13%) | 19 (0.04%) |
| Transplant rate (%) | 73.06 | 80.44[*] | 80.38 | 84.21 |
| Median wait (days)[a] | 61 | 503.5[*] | 500.5 | 655.0 |
| Wait from multi-list (days)[a] | – | 86.0 | 85.0 | 164.5 |
| Waitlist mortality (%) | 5.88 | 0.92[*] | 0.85 | 5.26 |
| Mean dist. between centers (NM)[b] | – | 379.2 | 372.9 | 731.6 |
| Max dist. between centers (NM)[b] | – | 2,256.6 | 2,254.7 | 2,256.6 |
| High urgency patients (%)[c] | 41.3 | 49.9 | 49.8 | 57.9 |
| Median patient status | 2 | 1 | 1 | 1 |

[*] $p < 0.001$ vs single-listed (Pearson's $\chi^2$ for rates, Mann-Whitney U for wait times).
[a] Median wait time calculated for transplanted patients only.
[b] For 3+ centers, distance is the maximum between any two centers.
[c] High urgency defined as status 1 (post-2018) or status 1A (pre-2018).

*Table A2.* Clinical outcomes of waitlisted patients for adult heart transplantation based on number of simultaneous center listings in the US (2010–2024). It is worth noting that while the median wait time for patients multi-listed at three or more centers is substantially higher (655 days) than for those at two centers (500.5 days), this difference may be an artifact of the small sample size. Furthermore, multiple listing demands significant time and additional medical evaluations across different centers, so healthier patients are better positioned to navigate these logistical hurdles and endure longer waiting periods.

| Listing strategy | Patients ($N$) | Transplant rate (%) | Med. wait (initial)[a] | Med. wait (multi)[a] |
|---|---|---|---|---|
| **Single-listed** | 49,013 | 73.06 | 57.0 | – |
| **Multi-listed** | | | | |
| Local (≤ 20 NM) | 267 | 76.03 | 464.0 | 92.0 |
| Regional (20–250 NM) | 415 | 79.28[**] | 494.0 | 131.0 |
| Long (250–1000 NM) | 269 | 83.64[**] | 607.0 | 79.0 |
| Very long (> 1000 NM) | 133 | 86.47[**] | 445.0 | 52.0 |

[a] Median wait time (days) for transplanted patients only.
[**] $p < 0.01$ compared to single-listed candidates (Pearson's $\chi^2$).
Note: long and very long distance candidates also have significantly higher rates than local candidates ($p < 0.05$).

*Table A3.* Impact of geographic distance between centers on transplant outcomes for adult multi-listed candidates in the US (2010–2024).

