# OpenReview forum: "Position: Machine Learning for Heart Transplant Allocation Policy Optimization Should Account for Incentives"
_ICML.cc/2026/Position_Paper_Track — ICML 2026 Position Paper Track spotlight_

### Official Review · Reviewer_pNWP · 2026-02-20

**Significance:** 3
**Argument Clarity:** 2
**Rating:** 4
**Confidence:** 2

**Questions:**

N/A

**Alternative Views Section:**

Yes

**Compliance With Llm Reviewing Policy A Conservative:**

Affirmed.

**Discussion Potential:**

2

**Final Justification:**

Given the positive response of the other reviewers and my lack of background with the topic I do not intend to block acceptance of the paper. I also appreciate the authors responses to my concerns. I will increased my score from 3 to a 4.

**Paper Summary:**

This paper argues that optimizing heart transplant allocation with machine learning must explicitly account for incentives and strategic behavior among stakeholders, because treating allocation as a static prediction/optimization problem risks policy failure in the face of strategic responses. The authors illustrate several incentive-related failure modes in today’s U.S. heart allocation pipeline and connect each to research directions spanning strategic classification, causal inference, mechanism design, and social choice.

**Position:**

Yes

**Position In Title:**

Yes

**Related Work:**

2

**Strengths And Weaknesses:**

My background is not in medicine, so please take my review with a grain of salt. I will follow the discussion with the other reviewers and am happy to adjust my score accordingly.


Strengths:

- Grounding with descriptive data and concrete examples, which helps with understanding and strengthens claims
- Good fit for the position track: it states a clear position, includes alternative views, and offers calls-to-action.


Weaknesses:

- Paper is often high-level. The connections to strategic classification/causal inference/mechanism design are plausible, but the paper could be clearer about which research problems are tractable now, what data access would be required, and what evaluation/benchmarking would look like in a policy setting
- Policy prescriptions risk unintended harm without guardrails. Suggestions like audits or discouraging certain behaviors could penalize clinical judgment or push behavior into harder-to-detect channels. The paper notes this as a potential issue, but does not offer a clear framework for preserving beneficial discretion while reducing harmful manipulation
- Scope/general-interest concern. The paper is tightly centered on US adult heart allocation with specific institutional details. Even if many incentive issues plausibly transfer to other organs or settings, the manuscript does not clearly distill the general ML problem statements and portable lessons (e.g., a generic evaluation protocol or reusable benchmark tasks) that would make the contribution broadly valuable to the wider ICML audience.

**Support:**

3

---

> ### Author Rebuttal · Authors · 2026-03-31
>
> We thank the reviewer for their service and valuable feedback.
>
> > Re. “Paper is often high-level. The connections to strategic classification/causal inference/mechanism design are plausible, but the paper could be clearer about which research problems are tractable now, what data access would be required, and what evaluation/benchmarking would look like in a policy setting”
>
> We believe that all of the concrete problems we highlight throughout the paper are, while challenging, technically tractable for the ICML community. Regarding data access, research in organ allocation relies either on the UNOS dataset, as we do, or on local registries from countries other than the US. Evaluation typically takes place in high-fidelity simulation (for example, using established transplant simulators). Certain mechanisms can also be validated through pilot studies before nationwide deployment. We should highlight that theoretically robust mechanisms perform well in practice, as we have seen time and again in mechanism design (for example, the success of spectrum auctions in the US). Finally, benchmarking should focus on direct comparisons against current status quo policies to measure the marginal improvement in incentive alignment and patient outcomes.
>
> > Re. “Policy prescriptions risk unintended harm without guardrails. Suggestions like audits or discouraging certain behaviors could penalize clinical judgment or push behavior into harder-to-detect channels. The paper notes this as a potential issue, but does not offer a clear framework for preserving beneficial discretion while reducing harmful manipulation”
>
> We agree with the reviewer: poorly designed interventions, such as audits, could have unintended consequences without guardrails. Developing a framework that preserves beneficial clinical discretion while reducing harmful manipulation is a challenge, and identifying solutions to this problem requires exactly the type of research agenda we call for.
>
> > Re. “Scope/general-interest concern. The paper is tightly centered on US adult heart allocation with specific institutional details. Even if many incentive issues plausibly transfer to other organs or settings, the manuscript does not clearly distill the general ML problem statements and portable lessons (e.g., a generic evaluation protocol or reusable benchmark tasks) that would make the contribution broadly valuable to the wider ICML audience.”
>
> While our analysis focuses on heart transplantation because of our collaboration with clinical heart experts, the issues we identify are not specific to heart transplantation. Heart transplantation serves as a representative case study for a broader class of ML problems: high-stakes resource allocation under strategic behavior. Every incentive misalignment we document arises in the entire organ allocation system. By providing a deep analysis into the specific domain of heart transplantation, our aim is to offer the ICML community a concrete set of challenges in developing incentive-aware ML models, which could be applied beyond organ allocation.
>
> We hope this addresses the reviewer’s concerns.

---

> > ### Author Rebuttal · Reviewer_pNWP · 2026-04-02
> >
> > Thank you for your rebuttal. Given the positive response of the other reviewers and my lack of background with the topic I do not intend to block acceptance of the paper. I also appreciate the responses to my concerns, and would recommend the authors to include the considerations on applications to different domains beyond heart transplantation into their paper. I will increase my score to a 4.

---

### Official Review · Reviewer_oTJU · 2026-03-08

**Significance:** 4
**Argument Clarity:** 2
**Rating:** 5
**Confidence:** 3

**Questions:**

On Causal Misalignment and Terminology: You argue that policy should "account for" incentives. However, your analysis of the 2018 policy change suggests that the policy’s own classification rules (the "means") actually created the strategic "device game" (the "incentive"). Given this, would it not be more accurate to position the research agenda as Incentive-Centered Design, where the goal is to build a structure that prevents the emergence of bad incentives, rather than a policy that merely reacts to them?

On the Ambiguity of "Policy": Throughout the paper, the term "policy" is used to describe both the algorithmic allocation rules—such as the 6-tier status system —and the broader regulatory monitoring systems, like the SRTR performance reports. Since these two layers have different objectives and stakeholders, how can a single "incentive-aware" machine learning model effectively optimize for both without creating new, unforeseen conflicts between clinical urgency and regulatory survival from the game theoretical perspective?

**Alternative Views Section:**

Yes

**Compliance With Llm Reviewing Policy A Conservative:**

Affirmed.

**Discussion Potential:**

3

**Final Justification:**

As the authors have fully addressed my concerns, I have raised my evaluation from 4 (borderline accept) to 5 (accept).

**Paper Summary:**

The paper argues that organ allocation is not merely a static optimization problem but a "complex game" involving multiple stakeholders who respond strategically to policy changes. Using US adult heart transplant data, the authors identify critical incentive misalignments—such as the "device game" to inflate priority and the strategic rejection of organs to protect performance ratings. They propose a research agenda integrating mechanism design and strategic classification to create "incentive-aware" policies.

**Position:**

Yes

**Position In Title:**

Yes

**Related Work:**

4

**Strengths And Weaknesses:**

The position paper presents a compelling argument that heart transplant allocation should be viewed as a game-theoretic mechanism rather than a static optimization task, primarily due to the strategic behavior of clinicians and centers who manipulate patient features to "game" priority. An empirical grounding provides definitive proof that clinical interventions are being used as signals to boost priority status. The authors effectively clarify the "cliff-edge" effects of current tier boundaries and propose continuous distribution as a solution to ensure that marginal changes in therapy result in only proportional priority shifts. Furthermore, granular analysis reveals critical stakeholder disparities, such as how large centers’ "offer abundance" leads to lower acceptance rates compared to small programs, and how multi-listing allows affluent candidates to exploit regional supply gaps.

However, a fundamental weakness of this position is a logic error regarding how policies are made: the title implies that "policy should account for incentives" as if incentives were an outside force, rather than recognizing that the way we build the system is what creates the incentives in the first place. This is worsened by the authors using the word "policy" to mean two different concepts—the allocation method used to rank patients (original meaning of policy) and the broader regulatory system used to evaluate hospital performance (related to incentives). While this confusion exists at the start of the paper, it is mostly cleared up in later sections. Another major concern is the "trust gap"; moving from simple rules to complex machine learning could create "black box" systems that doctors do not understand or trust, potentially leading them to ignore the algorithm's suggestions entirely.  While this is mentioned in the alternative view section, potential remedies for this problem are not sufficiently addressed.

**Support:**

3

---

> ### Author Rebuttal · Authors · 2026-03-31
>
> We thank the reviewer for their service and valuable feedback.
>
> > Re. “On Causal Misalignment and Terminology: You argue that policy should "account for" incentives. However, your analysis of the 2018 policy change suggests that the policy’s own classification rules (the "means") actually created the strategic "device game" (the "incentive"). Given this, would it not be more accurate to position the research agenda as Incentive-Centered Design, where the goal is to build a structure that prevents the emergence of bad incentives, rather than a policy that merely reacts to them?”
>
> We completely agree with the reviewer’s framing. When we state that policy should “account for” incentives, we do not mean a reactive, passive adjustment, but rather a design approach that centers on incentives. The research agenda we are advocating for is precisely about building the right incentive structures. We will add a clarification in the introduction to make our proactive stance more clear.
>
> > Re. “On the Ambiguity of "Policy": Throughout the paper, the term "policy" is used to describe both the algorithmic allocation rules—such as the 6-tier status system —and the broader regulatory monitoring systems, like the SRTR performance reports. Since these two layers have different objectives and stakeholders, how can a single "incentive-aware" machine learning model effectively optimize for both without creating new, unforeseen conflicts between clinical urgency and regulatory survival from the game theoretical perspective?”
>
> The reviewer correctly points out that “policy” in this domain often conflates allocation rules (the ranking algorithm, such as the 6-tier status quo) with regulatory oversight (for example, the SRTR performance monitoring). Our paper uses the term broadly to encompass the entire institutional mechanism, and we envision the development of multiple ML components to build the right incentive structures. It is necessary to co-design all these components so that the objectives align across the entire institutional mechanism. We will refine our paper to remove any ambiguity.
>
> > Re. “Another major concern is the "trust gap"; moving from simple rules to complex machine learning could create "black box" systems that doctors do not understand or trust, potentially leading them to ignore the algorithm's suggestions entirely. While this is mentioned in the alternative view section, potential remedies for this problem are not sufficiently addressed.”
>
> We appreciate this point. We agree that incentive-aware models must not only be technically robust but also transparent enough for clinicians to be able to verify them. We plan to expand our discussion on explainable ML and its relevance for healthcare. This challenge is another pressing research direction for the ICML community that applies to many high-stakes healthcare problems, not just organ allocation. Moreover, the medical community has developed practical methods to evaluate new systems before deployment, for example through dry runs, which have been successful in building trust for deploying algorithms for kidney exchange. That said, the modern kidney exchanges leverage autonomous matching use search/integer programming – the details of which are not understandable by medical doctors – over older, fully understandable dispatch rules. So, it is not always the case that the adopted solution is the more understandable one. Sometimes the community chooses efficiency over explainability, and kidney exchange is the prime example of that.

---

> > ### Author Rebuttal · Reviewer_oTJU · 2026-04-04
> >
> > I thank the authors for their detailed rebuttal and for addressing the concerns raised in my review. As my concerns (and questions) have been resolved, I have raise my evaluation by 1 point.

---

### Official Review · Reviewer_s9Gd · 2026-03-09

**Significance:** 3
**Argument Clarity:** 2
**Rating:** 4
**Confidence:** 3

**Questions:**

### Questions:

- In Section 2 (The device game), the authors mentioned that ML models should rely more on non-manipulable features. Could you briefly elaborate on what features are truly "non-manipulable" in a transplant setting?

- In Table A2, the median wait time for 3+ centers multi-listing is quite high (655 days) compared to 2 centers (500 days). Is this an artifact of the very small sample size, N=19, or does it reflect a specific clinical sub-population, e.g., highly sensitized patients? Adding a one-sentence footnote clarification would be helpful.

**Alternative Views Section:**

Yes

**Compliance With Llm Reviewing Policy A Conservative:**

Affirmed.

**Discussion Potential:**

2

**Paper Summary:**

This paper highlights a critical yet under-explored challenge in the US heart transplant allocation policy. The authors argued that the transition from rule-based allocation to ML-driven optimization typically ignores the strategic incentives of the stakeholders involved, including clinicians, hospitals, OPOs, and patients. They adeptly leveraged historical data UNOS 2010-2024 to expose several incentive misalignments across different stages of the decision-making pipeline, such as manipulating clinical features, exploiting opaque out-of-sequence allocations, risk-averse behavior induced by rigid performance monitoring, and multi-listing by affluent patients. To rectify these issues, the authors proposed an agenda urging the ML community to integrate strategic classification, mechanism design, and causal inference into future allocation policies, etc.

**Position:**

Yes

**Position In Title:**

Yes

**Related Work:**

3

**Strengths And Weaknesses:**

### Strengths

- The paper tackles a literal life-or-death resource allocation problem. It successfully demonstrates why machine learning will fail if deployed without incentive-awareness.

- The paper is well-structured, summarizing each identified bottleneck with specific and actionable recommendations.


### Weaknesses

- The authors rightly acknowledged that the surge in transplants around the spring SRTR reporting deadline is influenced by a "confluence of biological and logistical factors", e.g., seasonal mortality spikes or staffing limits during holidays. However, while the correlation to the SRTR deadline is intriguing, framing this strictly as strategic "horizon effect" gaming could be slightly overstated without a more rigorous causal inference framework to definitively prove the causal relations.

- This paper calls for incentive-aware policy optimization, but it does not address a critical practical question: how should ML researchers evaluate these policies before deployment? A brief discussion on how to build reliable, incentive-aware models or methods would add great practical value to the concluding remarks, which is currently missing.

- While the authors used the UNOS 2010-2024 dataset to support their claims, the manuscript seems to lack a detailed description of the data or some processing.

**Support:**

3

---

> ### Author Rebuttal · Authors · 2026-03-31
>
> We thank the reviewer for their service and valuable feedback.
>
> > Re. “The authors rightly acknowledged that the surge in transplants around the spring SRTR reporting deadline is influenced by a "confluence of biological and logistical factors", e.g., seasonal mortality spikes or staffing limits during holidays. However, while the correlation to the SRTR deadline is intriguing, framing this strictly as strategic "horizon effect" gaming could be slightly overstated without a more rigorous causal inference framework to definitively prove the causal relations.”
>
> We agree with the reviewer that this should be taken with caution, and we tried to emphasize that in the paper. We will make sure to revise the text so that it’s clear we are not drawing a definite conclusion, but rather making a hypothesis.
>
> > Re. “This paper calls for incentive-aware policy optimization, but it does not address a critical practical question: how should ML researchers evaluate these policies before deployment? A brief discussion on how to build reliable, incentive-aware models or methods would add great practical value to the concluding remarks, which is currently missing.”
>
> We thank the reviewer for bringing up this point. We argue that strong theoretical guarantees translate to robust and reliable performance for incentive-aware systems. We see this time and again in mechanism design. A notable example is spectrum auctions in the US, where well-designed auction formats led to massive improvements in efficiency. Moreover, the medical community has developed practical methods to evaluate new systems before deployment, for example through dry runs, which have been successful in building trust for deploying algorithms for kidney exchange. We will include these points in the concluding remarks to address the reviewer’s suggestion.
>
> > Re. “While the authors used the UNOS 2010-2024 dataset to support their claims, the manuscript seems to lack a detailed description of the data or some processing.”
>
> We provide the necessary information in the paper to reproduce our findings given access to the UNOS data, and we are happy to provide any additional information that the reviewer believes is relevant to our study. We should point out that given the sensitive nature of the dataset, it is not publicly available, but can be accessed through permission from UNOS. The UNOS registry is commonly used in prior research (including ML work) on organ allocation.
>
> > Re. “In Section 2 (The device game), the authors mentioned that ML models should rely more on non-manipulable features. Could you briefly elaborate on what features are truly "non-manipulable" in a transplant setting?”
>
> Features that capture biological factors, such as height, blood type and HLA sensitization, are harder to manipulate since they are physically determined and verifiable.  On the other hand, as we document in the paper, features that reflect clinical interventions (such as the decision to install a device such as LVAD or the use of specific inotropes) are more prone to manipulation since they depend on a human decision.
>
> > Re. “In Table A2, the median wait time for 3+ centers multi-listing is quite high (655 days) compared to 2 centers (500 days). Is this an artifact of the very small sample size, N=19, or does it reflect a specific clinical sub-population, e.g., highly sensitized patients? Adding a one-sentence footnote clarification would be helpful.”
>
> Thank you for the question. Yes, this is likely an artifact of the small sample size. Also, healthier patients are the ones that have the ability to multi-list and endure longer wait times and transport. For example, multiple listing across different centers requires significant time and additional medical evaluations. We will add this clarification in the revision.

---

> > ### Author Rebuttal · Reviewer_s9Gd · 2026-04-06
> >
> > Thank the authors for the responses. All of my concerns have been resolved.

---

### Official Review · Reviewer_2M2q · 2026-03-12

**Significance:** 4
**Argument Clarity:** 4
**Rating:** 5
**Confidence:** 3

**Questions:**

I’d have liked to see more discussion and summary on existing ML works addressing related problems in that stage, and evidence for the concrete areas that they fail to take into account the incentive misalignment.

**Alternative Views Section:**

Yes

**Compliance With Llm Reviewing Policy A Conservative:**

Affirmed.

**Discussion Potential:**

4

**Final Justification:**

I've gone through all the reviews and remain enthusiastic about the discussion potential this position paper can bring to the ML community and the direction of future research is may help open. I'll keep my original rating of Accept.

**Paper Summary:**

This paper presents the position that machine learning research in heart transplant allocation should account for incentives, and that organ allocation is not a static optimization problem but rather a complex game. To support this position, the paper highlights and explains incentive misalignment across several stages of the decision-making pipeline, and identify research directions for mitigating these vulnerabilities.

**Position:**

Yes

**Position In Title:**

Yes

**Related Work:**

3

**Strengths And Weaknesses:**

Strengths

This is a very interesting paper that gives a very comprehensive landscape of a research domain that could be of high relevance to the ICML community. It is likely to spark a spectrum discussions in the community.

This paper highlights a coherent central position (incentive misalignments) that are supported by clear evidence and background — the authors clearly identified and explained how these problems come about, and how they can affect the technical research in the various stages of organ allocation decision-making. These evidence reflect clear knowledge of the field and supported by clear literature.

Clear research priorities and directions are pointed out in response to the gaps identified.


Weaknesses

In each decision-making stage where the problem of incentive misalignment is elaborated, most of the space is dedicated to explain the background and provide evidence fo the presence of the problem. I’d have liked to see more discussion and summary on existing ML works addressing related problems in that stage, and evidence for the concrete areas that they fail to take into account the incentive misalignment.

**Support:**

4

---

> ### Author Rebuttal · Authors · 2026-03-31
>
> We thank the reviewer for their support, service, and valuable feedback.
>
> To address the reviewer’s suggestion, we will expand our discussion on how existing ML works deal with related problems and how they often fail to account for incentive issues.

---

> > ### Author Rebuttal · Reviewer_2M2q · 2026-04-02
> >
> > My comment was minor to start with. I'm not able to verify the revision, but I trust the authors will reflect the necessary changes in the revision.

---

### Decision · Program_Chairs · 2026-04-30

**Decision:**

Accept (spotlight)

**Comment:**

There was great discussion, where the authors persuaded reviewers.  The paper is extremely important and is written well.  It seems it should extend to other types of transplant as well, perhaps citing consortia focused on making these decisions more fair, see for example work of Lisa McElroy in liver transplant.